# In Vitro Release of Glycyrrhiza Glabra Extract by a Gel-Based Microneedle Patch for Psoriasis Treatment

**DOI:** 10.3390/gels10020087

**Published:** 2024-01-23

**Authors:** Ayeh Khorshidian, Niloufar Sharifi, Fatemeh Choupani Kheirabadi, Farnoushsadat Rezaei, Seyed Alireza Sheikholeslami, Ayda Ariyannejad, Javad Esmaeili, Hojat Basati, Aboulfazl Barati

**Affiliations:** 1Department of Biomedical Engineering, TISSUEHUB Co., Tehran 1956854977, Iran; ayehkhorshidian@gmail.com; 2Department of Tissue Engineering, TISSUEHUB Co., Tehran 1956854977, Iran; shrf.niloufar@gmail.com (N.S.); fateme.choopani@yahoo.com (F.C.K.); seyed_sheykholeslami69@yahoo.com (S.A.S.); reaalaydaa@gmail.com (A.A.); 3basati@gmail.com (H.B.); 3School of Pharmaceutical Sciences, Zhengzhou University, Zhengzhou 450066, China; 4Department of Biomedical Engineering, Faculty of Engineering, Islamic Azad University, Tabriz 54911, Iran; 5Department of Chemical and Biomedical Engineering, University of Missouri, Columbia, MO 65211, USA; frkb2@umsystem.edu; 6Department of Chemical Engineering, Faculty of Engineering, Arak University, Arak 3848177584, Iran; 7Department of Marine Biology, Faculty of Life Science and Biotechnology, Shahid Beheshti University, Tehran 1983969411, Iran; 8Tissue Engineering Hub (TEHUB), Universal Scientific Education and Research Network (USERN), Tehran 1956854977, Iran; 9Department of Chemical Engineering, Faculty of Engineering, Tehran University, Tehran 3584014179, Iran; 10Center for Materials and Manufacturing Sciences, Department of Chemistry and Physics, Troy University, Troy, AL 36082, USA

**Keywords:** microneedle patch, skin disorder, drug delivery, burst release, psoriasis

## Abstract

Microneedle patches are attractive drug delivery systems that give hope for treating skin disorders. In this study, to first fabricate a chitosan-based low-cost microneedle patch (MNP) using a CO_2_ laser cutter for in vitro purposes was tried and then the delivery and impact of Glycyrrhiza glabra extract (GgE) on the cell population by this microneedle was evaluated. Microscopic analysis, swelling, penetration, degradation, biocompatibility, and drug delivery were carried out to assess the patch’s performance. DAPI staining and acridine orange (AO) staining were performed to evaluate cell numbers. Based on the results, the MNs were conical and sharp enough (diameter: 400–500 μm, height: 700–900 μm). They showed notable swelling (2 folds) during 5 min and good degradability during 30 min, which can be considered a burst release. The MNP showed no cytotoxicity against fibroblast cell line L929. It also demonstrated good potential for GgE delivery. The results from AO and DAPI staining approved the reduction in the cell population after GgE delivery. To sum up, the fabricated MNP can be a useful recommendation for lab-scale studies. In addition, a GgE-loaded MNP can be a good remedy for skin disorders in which cell proliferation needs to be controlled.

## 1. Introduction

The ongoing uncontrolled multiplication of cells is the primary aberration that leads to the emergence of illnesses. The symptoms and severity of skin conditions vary widely. They may be fleeting or long-lasting, painful or not. Some have environmental causes, while others could have genetic causes [1]. While most skin disorders are minor, others can indicate a more serious problem. There are diseases that are caused by changes in the cell population in the skin. Eczema, psoriasis, acne, rosacea, ichthyosis, vitiligo, hives, seborrheic, and dermatitis are the most common skin disorders [2].

Skin architecture and function depend on diverse populations of epidermal cells and dermal fibroblasts. Multiple epithelial stem cell (SC) populations have been shown to contribute to skin homeostasis [3]. Physical parameters such as tension, pressure, or temperature contribute to the cellular microenvironment, are sensed by distinct cell populations, and influence cellular fate [4]. “Proliferative skin disease” is a phrase that we define to include all skin diseases characterized by an abnormal proliferation of epidermal keratinocytes [5]. Although these proliferative diseases (e.g., psoriasis, cancer, eczema, and ichthyosis) have multiple causes, the universality of common physiological mechanisms means that the molecular basis of cell proliferation has mechanisms common to all cells [6].

The primary causes of psoriasis are understood to be excessive keratinocyte proliferation and aberrant keratinocyte differentiation, as well as infiltration of many inflammatory cells [7]. However, in most cases, the first prescribed treatment will be a topical treatment, such as vitamin D analogs or topical corticosteroids, creams, and ointments applied to the skin. If these are not effective, phototherapy may be prescribed [8]. By the way, controlling the abnormal proliferation of cells seems to be a useful mechanism in controlling or healing skin disorders.

Herbal remedies have been used in traditional medicine practices for centuries for a wide range of diseases and are a promising alternative, offering a substantial improvement in patient conditions and significantly decreasing disease symptoms [9]. Investigating the proliferation, differentiation, and cytotoxic effects of different herbal extracts on stem cells may provide in-depth insights into their disease-curing mechanisms. However, herbal extracts have shown much promise for cell proliferation and differentiation in many different studies [10,11].

Herbal extracts contain a plethora of phytochemicals such as polyphenols, flavonoids, and other plant-derived chemicals that synergistically aid in treating diseases in traditional medicine methods [10]. As these extracts are composed of naturally occurring medicinal herbs, which may be regularly consumed by local communities, these may cause the least side effects and have lower toxicity than current stimulants [12].

Medical devices with a micron scale called microneedle patches are used to administer drugs. To give vaccinations, medications, and other therapeutic substances, they are made up of small needles that pierce the skin. The usage of microneedles has expanded beyond transdermal drug administration to include intraocular, vaginal, transungual, cardiac, vascular, gastrointestinal, and intracochlear delivery [13]. Small molecular medications, macromolecular pharmaceuticals (proteins, mRNA, peptides, vaccines), and hydrophilic or hydrophobic drugs can all be delivered via microneedles.

There are several types of microneedle patches used for drug delivery and other applications including solid microneedles, coated microneedles, dissolving microneedles, hollow microneedles, and hydrogel microneedles. It is vital to remember that depending on the individual application and preferred drug delivery mechanism, several sorts of microneedles may have distinct benefits and disadvantages. To increase the effectiveness of medication administration and the patient experience, researchers are still investigating and creating new kinds of microneedle patches.

The main aim of this study was to evaluate the potential of the obtained herbal extract from the root of *Glycyrrhiza glabra* in controlling the cell population using a new low-cost fabricated microneedle patch (MNP). This study was conducted as an in vitro study to figure out if it is possible to slow down the cell proliferation rate, which can be a good candidate to treat appropriate skin disorders.

## 2. Results and Discussion

### 2.1. HPLC Analysis

Figure 1A shows the results of the HPLC analysis for the Gg extract. It is clear that some peaks existed in the extract that can be attributed to the different components in the extract. The most important one is related to glycyrrhizic acid (known as Enoxolone (ENx)). It has been discovered that Enoxolone is a pentacyclic triterpenic organic acid [14]. Licorice root also contains other phytocompounds such as glycyrrhizin, 18β-glycyrrhetinic acid, glabrin A and B, and isoflavones [15]. However, it was also reported that Enoxolone can be extracted from biological materials including rat blood, human urine, and rat bile [14]. It can be concluded that this agent exists in the human body and no side effect can be hypothesized for its release under transdermal delivery approaches. The primary active component of licorice root, which has been used for centuries as a sweetener and traditional herbal remedy, is glycyrrhizic acid. Asthma and arthritis are inflammatory disorders, and glycyrrhizic acid has been demonstrated to have anti-inflammatory properties. It has been discovered to have direct antiviral action against SARS-CoV-2 and the hepatitis virus. Glycyrrhizic acid may be used to treat cancer due to reports that it possesses anticancer properties. Given that glycyrrhizic acid has been demonstrated to have neuroprotective properties, it may be used to treat neurological conditions including Parkinson’s and Alzheimer’s. Asthma and hay fever are two allergic illnesses for which glycyrrhizic acid has been reported to have antiallergic benefits. The results from the HPLC confirm the presence of this ingredient. The study by Trupti W. Charpe et al. proves the presence of glycyrrhizic acid as the main ingredient [16].

### 2.2. Fabricated Patch 

In this study, a simple, low-cost, and easily fabricated MNP was introduced. The fabricated microneedles were similar and nearly all had a diameter of 400–500 μm and a height of 700–900 μm. This variance in the dimensions can be attributed to the power of the CO_2_ laser cutter during working and enhancing the temperature. Figure 1B demonstrates the GgE-loaded MNP and, as can be seen, all have a conical shape with an acceptable sharpness. However, previous studies have reported several types of microneedles with a distinct sharpness [17,18,19,20]. The light brown color of the microneedles confirms the presence of the GgE. The tip of all the microneedles looks dark, which can be justified by the accumulation of the GgE. This issue is hypothesized to be useful for a better and deeper release of the GgE. Compared to the previous studies, in this study, acrylic sheets were employed, while they used Poly dimethyl silicone, which resulted in more polymer diffusion into the cavities [21]. 

### 2.3. Structure of Microneedles

Figure 1C shows the microscopic images of the fabricated GgE-loaded MNP. Based on the SEM images, it can be concluded that the microneedles were conical with minimum roughness and a lack of porosity. This can prove the dense structure of the microneedles with no trapped air. The sharpness of the created microneedles seems to be the same. It also shows that the surface of the microneedles is smooth with tiny particles, which can be attributed to the deposition of the GgE.

### 2.4. Microneedle Penetration

For the evaluation of the microneedle penetration efficacy, compared to the previous studies, the GgE-loaded MNP could penetrate into the parafilm and also create holes (Figure 2A). The created holes were circular (200–250 μm in diameter), which confirms the microneedles’ good mechanical strength to tear the parafilm. The penetration was also tested on real human skin (two soft and hard parts) (Figure 2B), and based on our visual evaluation, the created microneedles could rip the skin and leave holes (blood was observed in some points). It is also important to highlight the MNP failed in some areas and the reason can be attributed to the nonuniform pressure loading or low mechanical strength of needles in that zone. The question is how much penetration the MNP made through the skin, the answers to which can be found by more in vivo and pathological studies (outside of the main aim of our study).

### 2.5. Rhodamine Delivery

Because MNPs are intended to be a substitute for conventional needles, the drug release efficacy is a crucial concern. There are numerous publications on the potential of chitosan-based MNPs for the release of various biomolecules (nicely reviewed by Prausnitz et al., [18]). Regardless of the drug, a fundamental challenge in the fabrication of microneedles is to ensure that there is a significant drug load in the tiny needles. Therefore, ensuring drug release is necessary for microneedle designing. Figure 3A demonstrates the rhodamine release from the rhodamine-loaded MNP in a sodium alginate layer (considered as a semi-skin layer) under a continuous water flow for 30 min. It is necessary to mention that this layer just plays a role as a reservoir to test the drug release from the MNP. Certainly, to make a layer mimicking real skin, it needs more experiments and study to evaluate it from distinct aspects including the tensile strength, water content, degradation, porosity, and thickness. After 15 and 30 min, the MNP showed a notable rhodamine release into the SLD matrix. It depicts that the chitosan MNP had the potential for a burst release of rhodamine after 15 and 30 min. However, the size of the loaded biomolecule is critical in release efficacy [22]. Water uptake and swelling are the main features that can affect drug release because due to the variance in the drug concentration (in the MNP and tissue), water is replaced with the drug and drugs diffuse out from the MNP [23]. To make this issue more clear, the rhodamine-loaded MNP is dry, while the SLD is saturated with water, and after inserting the MNP in the SLD, the microneedles start water adsorption and swelling. The penetrated water dissolves rhodamine and due to the difference in rhodamine concentration inside and outside of the microneedles, rhodamine diffuses out of the microneedles (inside the SLD). In the case of insertion in human skin, a similar mechanism happens due to the skin interstitial fluid. 

### 2.6. Swelling and Degradation

Figure 3B shows the swelling behavior of the GgE-loaded MNP every minute for 5 min, and then every 5 min for 20 min. It confirms the swelling behavior of the microneedles and based on Figure 3C, an increasing trend in swelling can be seen till 15 min later. As can be seen, 1 min after inserting the MNP, the microneedles are swollen (0.56%), which means after 1 min, GgE is expected to be released. After 5 min, the swelling ratio enhanced to 1.6%. During this time, it is obvious that the microneedles are swollen and due to this phenomenon, they are losing their mechanical strength, so it is recommended to never remove the patch from the skin until the specified time. Finally, after 10 min, it reached around 2%. After 15 min, no significant difference was observed compared to t = 10 min. After 15 min, no swelling was observed, so no data were reported at these intervals (Figure 3C). An interesting point is that the substrate did not swell during the first 5 min, but after that (t = 10 min), its deformation due to the swelling was obvious. 

The effect of swelling on drug release can be significant and should be carefully considered in pharmaceutical formulations. When a drug delivery system comes into contact with a liquid, it may undergo swelling due to the absorption of the liquid by the polymer matrix. This swelling can affect the release of the drug from the delivery system. One important aspect to consider is the swelling kinetics. Different polymers have different swelling characteristics, and their ability to take up liquid can vary greatly. It is crucial to understand the kinetics of swelling for the specific polymer used in the drug delivery system. This can be determined experimentally by measuring the change in the size or weight of the polymer as it absorbs the liquid [24]. When the microneedles penetrate the skin, they induce a local immune response, leading to swelling. This swelling can potentially affect the release of drugs from the microneedles. Moreover, microneedle swelling can also modify the microenvironment around the microneedles. The increased swelling can lead to changes in the local pH, temperature, or moisture content. These changes can influence the solubility and stability of the drug molecules, further impacting their release [25]. It is important to note that the effect of microneedle swelling on drug release can vary depending on several factors, including the composition of the microneedles, the properties of the drug, and the specific application [26].

According to the observation, microneedles begin to degrade after 15 min, which causes them to become shorter. After 25 min, it is possible to assert that the microneedles have completely degraded, at which point it can be assumed that a 100% release has been accomplished. The main justification for the fast swelling and quick degradation of microneedles can be the lack of a crosslinking step in the MNP fabrication process. This design is suitable for drugs that must have a burst release. In contrast, this design can be modified for biomolecules that must be injected gradually [19,27]. (Note: Based on Figure 3B, it is observed that the patch substrate is swollen. This is due to the contact between the substrate and the SLD, which is saturated with water. As for the skin, the substrate is dry when it is in contact with the skin and does not swell, and only the microneedles penetrate the skin(. Because the size of the swollen microneedles is greater than the diameter of the hole that has been made in the skin, or the likelihood of the microneedles breaking during removal of the patch is increased, this swelling behavior can be useful in holding the patch on the skin without any external pressure. 

Understanding the many elements that may cause medication degradation in microneedle patches is crucial. These variables include, among others, the temperature, pH, and humidity. In terms of drug release, degradation can significantly impact the release profile of the drug from the patches [28]. As the polymeric matrix degrades, the polymeric networks may change, which can affect its solubility and release kinetics. Therefore, microneedles gradually release drugs during their slow degradation under the skin. It highlights the point that the performance of a microneedle patch can be affected by the degradation of the carrier matrix, which is directly linked to drug diffusion, dissolution, and degradation.

### 2.7. GgE Release and Kinetic Study

The quantitative release of GgE from MNPs has been one of the main concerns in this study. Although GgE exists in both the microneedles and substrate, generally, the release is going to start from the microneedles. However, GgE diffusion from the substrate into the microneedles and then into the SLD matrix cannot be neglected. By the way, by using a microfluidic device, monitoring the correct amount of drug release from the MNP with a diameter of 1 cm with 52 microneedles was tried. It was assumed that the amount of released GgE from the substrate was ignorable. Furthermore, it has been agreed that the release study should last for around 45 min based on the findings of the swelling analysis, which showed that the substrate virtually began swelling after 15 min.

Figure 3D demonstrates the standard curve to determine the exact concentration of the released GgE. Figure 3E depicts the UV spectrum of the samples taken at different intervals. The graph has been plotted based on time (h) and accumulation release. Based on the results, no GgE was detected until 5 min. An amount of 6 μg/mL of GgE was detected at t = 5 min. The concentration of the released GgE at t = 10 min reached 8 μg/mL, and after 15 and 20 min, it reached 0.11 μg/mL and 19 μg/mL, respectively. By prolonging the process until 25 and 30 min, the concentrations of GgE were reported equal to 26 and 34 μg/mL, respectively. Finally, after 45 min, specific changes occurred in the GgE content (0.51 μg/mL), which can be due to the swelling of substrates. These results are in accordance with that swelling.

Figure 4 depicts the fitting between the experimental results and data from the models. Considering the zero-order model, there was a good fit between the experimental and model results with R^2^ equal to 0.987 and the *K* value equal to 1.247.
*F = Mt/M0 = Kt = 1.247t*(1)

In the case of the first-order model, there was no notable fitting and R^2^ was not in an acceptable range. The Korsmeyer–Peppas model showed that it can be a good fit for the experimental results with R^2^ equal to 0.992. This model has two main parameters, known as the constant *K* value and n as the power of time. These values were equal to 1.460 and 1.254, respectively.
*F = Mt/M0 = Kt^n^ = 1.46t*^1.254^(2)

The Higuchi showed a close fitting compared to the Korsmeyer–Peppas model, but in general, it was not accepted as a good fitting model. R^2^ was equal to 0.947. The Higuchi is similar to the Korsmeyer–Peppas model with *n* = 0.5. The *K* value for the Higuchi model was equal to 0.811.
*F = Mt/M0 = Kt^0.5^ = 0.811t*^0.5^(3)

In general, it can be claimed that the release kinetic for the GgE-loaded patch follows the Korsmeyer–Peppas model. The Korsmeyer–Peppas model is a widely used mathematical model for describing drug release from hydrogels. It is based on the assumption that the drug release follows a diffusion mechanism. In this model, the release rate is related to time (*t*), the release exponent (n), and other parameters such as the initial drug loading and the geometry of the hydrogel matrix. In this model, *n* < 0.43 indicates Fickian diffusion, 0.43 < *n* < 0.85 indicates relaxation-controlled, and *n* > 0.85 indicates non-Fickian (anomalous) transport [29]. In our study, *n* = 1.461, from which it can be claimed that the GgE release follows non-Fickian diffusion.

### 2.8. Cell Viability

Previous studies reported the high biocompatibility of chitosan as scaffolds [30], drug delivery systems [31], wound dressing [32], nanocomposites [33], and even MNPs [34]. Chitosan is biodegradable and biocompatible, and its immune-stimulating activity can increase both cellular and humoral responses. The fabricated MNPs in this study were evaluated after crosslinking the MNPs by TPP (0.25 mg/mL) and washing them using sterile PBS. Crosslinking was performed to prevent the fast degradation of chitosan MNPs during cell culture for 48 h. Figure 5A provides the biocompatibility of the pure MNP (without GgE) against the L929 cell line after 24 h, 48 h, and 72 h. As well as the previous studies, high cell viability was observed for the fabricated MNPs.

The impact of GgE and its delivery in the matrix of the SLD using the GgE-loaded extract was carried out according to Figure 5B. In this technique, the control group received no GgE, while the treated group was exposed to GgE for 5 min after 12 and 24 h. Based on the release analysis, it can be predicted that cells receive nearly 6 μg of GgE per each MNP administration. Biological staining is a good candidate to check the number of living cells. Changes in the cell population have been monitored via DAPI staining (Figure 5C) and acridine orange (AO) staining (Figure 5D).

One of the purposes of DAPI staining is to study the cell cycle, determine the index of mitosis in an organism, or count cells [35]. The DAPI staining results revealed that both groups’ cell populations expanded properly, with distinct cell nuclei and no discernible dead cells (Figure 5C). In the control group, the cells seemed to be attached to the SLD and with a higher concentration after 48 h in contrast to 24 h. Based on Figure 5C, the GgE delivery reduced cell proliferation compared to the control group. This finding indicates that in skin disorders like Psoriasis, in which cells have non-stop proliferation, a transdermal delivery system like the GgE microneedle patch can reduce the growth of cells and control this disease. 

Comparing the control group with the treated group (Figure 5(Di) and Figure 5(Dii)), it can be seen that the rate of cell proliferation was slower when the GgE was released. Comparing the AO results at different intervals depicted that the presence of the GgE slowed down cell proliferation (Figure 5D).

There is a question about how GgE could lower the rate of cell proliferation. More biological testing is required to provide a mechanism to answer this question, and finding its answer was beyond the major focus of this study. According to our analysis of the literature, ENx, also known as glycyrrhetinic acid and derived from the herb licorice, is one of the key components of GgE. It has been reported that it has been found to induce G1-phase cell cycle arrest in human non-small-cell lung cancer cells through the endoplasmic reticulum stress pathway [36].

Based on earlier findings, the ENx protective activity was assumed by exerting anti-inflammatory, anti-catabolic, oxidative stress-decreasing effects, gastroprotective, antiviral, cardioprotective, anti-tumor, neuroprotective, and hepatoprotective activities in animal models [37]. ENx can be used as a flavoring agent in food and to mask the bitter taste of drugs such as aloe and quinine. ENx has also been found to modulate vascular injury and atherogenesis. 

The ENx anti-inflammatory effects are based on an inhibitory effect on neutrophils, which are the main mediators of inflammation, producing superoxide radicals. It also provided anti-inflammatory effects by reducing the levels of inflammatory nitric oxide, prostaglandin E2, and intracellular reactive oxygen species and by suppressing the expression of pro-inflammatory traits by inhibiting NF-κB and phosphoinositide-3-kinase activity [38]. Furthermore, it has antibacterial and antifungal properties due to its steroid-like structure. 

For its firming, moisturizing, whitening, and antiaging actions, ENx is utilized in skin cosmeceuticals to maintain the health and condition of the skin. Recent studies have revealed that ENx and its derivatives have some anticancer effect against a variety of cancer cells and can also cause cell death [39]. Notably, these compounds were found as potent inhibitors of transcription factor NF-κB. NF-κB is a transcript factor discovered by Sen and co-workers in 1986, which is usually overexpressed and constitutively activated in many types of malignancies [37]. The 1–4 NF-κB, as a family of related protein hetero- or homodimers, promotes the downstream protein expression of anti-apoptosis (XIAP, Bcl-2, and Bcl-xL), proliferation (c-Myc) and invasion (MMP-2 and MMP-9), and exhibits remarkable capabilities for regulating the transcription of hundreds of target genes [40]. 

Increasing evidence indicated that the aberrant activation of NF-κB pathways allows cancer cells to escape apoptosis, invasion, and metastasis, which contributes to cancer drug resistance. Because activation of NF-κB is an essential feature of the survival of cancer cells during treatment, which results in treatment resistance, considerable researchers have focused on targeting NF-κB for cancer therapy [41]. 

ENx inhibition of TNF-α-activated JNK/c-Jun and IκB/NF-κB signaling pathways, which regulate, respectively, AP-1- and NF-κB-mediated gene transcription, contributes to the suppression of ICAM-1 (playing a key role in the early stage of inflammatory response) [42]. NF-κB regulates TNF-α stimulated ICAM-1 expression at the transcriptional level in vascular endothelial cells. Its movement is interceded by homodimeric or heterodimeric combinations of NF-κB family proteins, such as p50, p65, and c-Rel. In its resting state, NF-κB is present in association with its cytoplasmic inhibitor, IκB [42]. Once activated by an inflammatory cytokine, such as TNF-α, IκB was rapidly phosphorylated and degraded, leading to the translocation of activated NF-κB from the cytoplasm to the nucleus24. ENx observably inhibits IκB degradation and NF-κB p65 translocation and subsequently reduces NF-κB DNA binding activity, recommending a decrease in ICAM-1 expression by ENx by blocking the classic inflammatory pathway of the IκB/NF-κB system. This reduced expression may, at least in part, account for the mechanism of ENx exerting its anti-inflammatory effects. Furthermore, ENx as well as NF-κB and JNK inhibitors suppress TNF-α-induced ICAM-1 protein expression, suggesting that NF-κB and AP-1 are important in regulating cytokine-induced ICAM-1 expression. In addition, ENx inhibition of ICAM-1 is mediated by its down-regulation of NF-κB and JNK signaling pathways [42,43].

## 3. Conclusions and Future Prospective

The main backbone of this study comprised addressing two issues: firstly, recommending a low-cost MNP fabrication for in vitro studies. Secondly, the evaluation of the GgE impact on slowing down the cell proliferation rate. Based on the results, the fabricated microneedles were conical and sharp enough, and showed notable penetration into parafilm, hydrogel, and skin. The release of GgE also showed good control over the cell population. However, more biological studies are recommended to prove the potential of GgE to heal skin disorders like psoriasis.

It may be a good idea to use GgE for in vivo investigations and other drug delivery methods in the future. There may be additional justifications for introducing GgE as a herbal treatment for skin disorders/diseases if cell cycle and cell signaling pathways are studied. 

Understanding the main mechanism can result in new ideas to optimize the concentration and also develop new delivery systems.

Employing a new MNP with additional microneedles and various diameters in this situation may provide greater difficulties.

## 4. Materials and Method

### 4.1. Materials

Sodium alginate (medium viscosity, Sigma, Saint Louis, MO, USA), chitosan (low molecular weight, ≥75% deacetylation, Sigma), Thiazolyl Blue Tetrazolium Bromide (Sigma), and sodium triphosphate (TPP, Merck, Rahway, NJ, USA) were purchased from a local supplier (Alborz Shimi Co., Tehran, Iran). Phosphate-buffered saline, 4,6-diamidino-2-phenylindole dihydrochloride (Invitrogen, Waltham, MA, USA), acridine orange ethidium bromide (Invitrogen) were provided by KalaZist Co. (Tehran, Iran). All reagents were analytical grades.

### 4.2. Extraction

An amount of 10 g of the air-dried root of the *Glycyrrhiza glabra* plant was mixed with 90 mL of 90% hydroalcoholic solvent under stirring (500 rpm) at 60 °C for 24 h. The *Glycyrrhiza glabra* extract (GgE) was concentrated using a rotary evaporator under reduced pressure at 40 °C. Next, the obtained viscose solution was lyophilized using a freeze-dryer (DORSA TECH CO., Iran, Alborz). The final concentration was equal to 0.06 g/mL of extract. The final product was analyzed by HPLC (Agilent 1260 series, Santa Clara, CA, USA). The employed column was an Eclipse XBD-C18 reversed-phase column (250 mm× 4.6 mm, 5 m). The mobile phase consisted of a mixture of 30% acidified water (1% acetic acid) and 70% methanol. The flow rate was maintained at 1 mL/min.

### 4.3. Preparation of GgE-Loaded MNP

#### 4.3.1. Microneedle Mold Fabrication

The preparation of the MNP mold was performed according to the instructions reported by Rezaei Nejad with a little modification (Figure 6A) [21]. Briefly, acrylic disks with a diameter of 15 mm were created using a CO_2_ cutter (Vera) with a maximum power of 100 W. Then, using the CO_2_ laser, the laser beam was adjusted on the surface of the acrylic disk. The laser power and pulse width were the main parameters to be optimized for desirable engraving and almost the same depth per run. The engraving was performed by a simple laser shot (17 W and 5 millisecond) on the selected points designed by CorelDraw. The depth of the created cone could be easily adjusted by changing the power and pulse width. Power is a crucial laser parameter since it determines the quantity of energy that is used throughout the procedure and maybe even how deep the cut will be. The time interval between a single laser pulse’s beginning and ending is referred to as the pulse width. Higher cutting speeds and improved laser cutting quality are made possible by using the proper pulse widths. The ideal pulse width varies depending on the workpiece material. Increasing the power and the rate of engraving resulted in a too-depth cone (>1 mm). The fabrication process continued to complete the mold. The engraved acrylic mold was washed with isopropanol several times and then rewashed with deionized water to remove the dust and particles from the mold. The mold was vacuum-dried for 30 min. An acrylic cylinder with an inner diameter of 10 mm and an outer diameter of 15 mm was also fabricated using the CO_2_ laser cutter. A plastic washer (for sealing) with the same diameter (according to the cylinder) was also fabricated. This plastic washer was placed on the main microneedle mold and then, the cylinder was located on the plastic washer (Figure 6B). 

#### 4.3.2. Preparation of Chitosan/Extract Solution

An amount of 1 g chitosan powder was gradually dissolved in 9 mL 1% *v/v* acetic acid at room temperature (200 rpm) for 6 h. Then, the unreacted acetic acid was removed using a dialysis bag (12 KD) in distilled water overnight. An amount of 100 mg of the dried GgE was dissolved in the final purred chitosan solution and mixed for 2 h at RT. The final chitosan/extract solution had a concentration of 10 mg/mL. The obtained solution was transferred into a 5 mL syringe and was left to remove the bubbles. Finally, the syringe was stored in a refrigerator at 10 °C.

#### 4.3.3. Preparation of MNP 

Fabrication of degradable microneedles was carried out using the chitosan/extract solution (10 mg/mL) according to Figure 6C. Initially, the acrylic mold was exposed to oxygen plasma for 5 min to create a more hydrophilic surface for better diffusion of the chitosan/GgE solution into the cavities. Frequently, 0.5 mL of the chitosan/GgE solution was dripped over the mold and allowed to cover the surface of the mold. Then, the mold was centrifuged (using specific holders) at 4000 rpm for 10 min. The centrifuge plays a vital role in the fabrication of the patch with sharp microneedles. Next, the mold was dried at 40 °C overnight. Finally, the patch was detached and stored at 4 °C.

### 4.4. SEM

To study the morphology and dimension of the created microneedles, SEM analysis (scanning electron microscopy) was performed using Philips XL30 (Philips, Amsterdam, The Netherlands, 25 kV). The patch was coated with a layer of gold using a spotting coater [44].

### 4.5. Semi-Skin Preparation

To prepare a skin-like disk (SLD), 1 g of sodium alginate was dissolved in 9 mL of water to prepare a 10% sodium alginate solution. Then, the solution was poured into cylindrical molds with an inner diameter of 10 mL and a depth of 2 mm. The molds were then frozen at −20 °C for 24 h. Next, the freeze-dried sodium alginate disks were immersed and crosslinked in a 10% w/v calcium chloride solution for 30 min at 10 °C. Finally, the crosslinked sodium alginate disks were washed and freeze-dried. The final SLDs were stored in the refrigerator. 

### 4.6. Penetration Test

Following the recommended protocol by Rezaei Nejad et al., with a little modification, parafilm was nominated to test the penetration of the MNP. A piece of parafilm in 2.5 × 2.5 cm^2^ was prepared. The patch was placed on the film and pushed by hand. This test was repeated three times and the number of created holes was counted [21].

### 4.7. Release Analysis

*Color tracing:* To study the efficacy of the patch in the delivery of pharmaceutical agents, a rhodamine-loaded patch was fabricated according to Figure 6C. A microfluidic device according to Appendix A was employed [45]. The system comprised a mini pump, reservoir, chip, and cap. The chip was designed simply and made of a membrane, one straight microchannel, and a circular zone (diameter:12 mm, depth: 4 mm). The membrane was placed at the bottom of the circular zone and the microchannel passed under the membrane. The SLD was placed on the membrane. Finally, a rhodamine-loaded patch was placed on the SLD and pushed (to make sure that the needles were penetrated in the SLD matrix), and sealed by the cap. This system provides enough moisture in the SLD and due to the difference in the concentration of rhodamine in the patch and SLD, rhodamine is supposed to diffuse into the SLD. After 15 and 30 min, the SLD was checked for diffused rhodamine.

*GgE tracing:* Using the same approach (microfluidic system), GgE-loaded patches were located in the microfluidic device, and sampling was performed from the reservoir (1 mL) at different intervals (2, 5, 10, 15, 20, 25, 30, 45 min). Using a UV–vis spectrophotometer, the adsorption of the sample was measured in the range of 200–400 nm. After measurement, the sample was added to the reservoir.

The final release profile was studied to find out the most suitable model based on Korsmeyer–Peppas, Higuchi, first-order, and zero-order models. 

### 4.8. Swelling

Swelling of the microneedles plays a vital role in drug delivery and drug diffusion. To aim for this, a GgE-loaded patch was pushed on a swollen SLD. After t = 1, 2, 3, 4, 5, 10, 15, 20, and 25 min, the patch was removed and analyzed visually. The width of the swollen microneedle was measured using ImageJ software (version 1.52v). The degree of swelling was calculated by Equation (4):(4)$$\%\;swelling=\frac{d_{2}-d_{1}}{d_{1}}\times 100$$

In which, *d*_1_ and *d*_2_ are the diameters of the microneedles at t_0_ and t.

### 4.9. Biodegradability

The degree of degradation was measured at RT based on the disappearance of the microneedles by analyzing the height of microneedles using the ImageJ software (version 1.52v). After t = 1, 2, 3, 4, 5, 10, 15, 20, and 25 min, the patch was removed and analyzed visually. During degradation, the height of the microneedle was decreased. The degree of degradation was calculated by Equation (5). This approach seems to be more reliable than measuring the weight of the whole patch.
(5)$$\%\;Degradation=\frac{h_{1}-h_{2}}{h_{1}}\times 100$$

In which, *h*_1_ and *h*_2_ are the diameters of the microneedles at t_0_ and t.

### 4.10. Biocompatibility 

The patches were sterilized by immersion in 70% ethanol for 24 h and then, by UV rays (254 nm) for 1 h. The patches were then washed several times with sterile PBS. Then, the patches were seeded with 1 × 10^4^ fibroblast cell line L929 cells/well in a 96-well plate. Next, the patches were incubated for 48 h at 37 °C and 5% CO_2_. After 48 h, 10 μL of the MTT reagent (0.5 mg/mL) was added to each well and incubated again for 4 h. Afterward, 100 μL of the solubilization solution was added to each well. The plate was then left for incubation overnight. The cells in the wells without patches were treated as the positive control. The purple formazan crystals were checked, and the absorbance was measured at 540 nm by a Wareness Technology Microplate-Reader. All experiments were repeated three times [46].

### 4.11. Efficacy of GgE Delivery on Cell Population 

To evaluate if the release of GgE via an MNP can slow down cell proliferation, the following approach was performed according to ISO 10993-5. Firstly, the SLD, which was a hydrogel, was washed with sterile PBS. Then, it was seeded with 1 × 10^4^ L929 cell/well in a 24-well plate and incubated for 12 h at 37 °C and 5% CO_2_. After 12 h, a sterile GgE-loaded patch was pushed on the cell-seeded SLD and incubated for 5 min. During this time, it was predicted that GgE is released and diffused into the cell-seeded SLD. After 5 min, the patch was removed and the cell-seeded SLDs were incubated for 24 h. Again, another sterile GgE-loaded patch was pushed on the cell-seeded SLD and incubated for 5 min. After 5 min, the patch was removed and the cell-seeded SLDs were incubated for 24 h. The SLDs were analyzed by DAPI (4′,6-diamidino-2-phenylindole) staining and acridine orange staining after 12, 24, and 48 h.

#### 4.11.1. DAPI Staining

To make sure about the cell proliferation, DAPI staining was performed. With 1% PBS, the SLDs were washed for 5 min (repeated two times). The SLDs were transferred into a 4% paraformaldehyde solution for 10 min and frequently washed with PBS. Then, 0.1% Triton solution was added to each sample. After 5 min, the SLDs were rewashed with PBS. Next, 150 λ of 4, 6-diamidino-2-phenylindole dihydrochloride (10 µg/mL, DAPI, Sigma, as a nuclear stain) was dripped over the SLDs for 5 min. Then, the SLDs were rinsed three times with PBS. The stained SLDs were kept in the dark and PBS until fluorescence microscopy.

#### 4.11.2. Live–Dead Assay

Acridine orange staining was carried out for the SLDs after 12, 24, and 48 h. Dual fluorescence staining solution containing 100 μg/mL acridine orange ethidium bromide (Sigma Aldrich, St. Louis, MO, USA) was added to samples and washed with PBS after 5 min. The living L929 cells were imaged using a fluorescence microscope (LM, Leica 090-135002, Wetzlar, Germany).

### 4.12. Statistical Analysis

GraphPad Prism software (version 12) and ANOVA analysis were used to statistically analyze the results. The results were expressed as mean ± standard deviation, and *p* < 0.05 was considered a significant difference.

## Figures and Tables

**Figure 1 gels-10-00087-f001:**
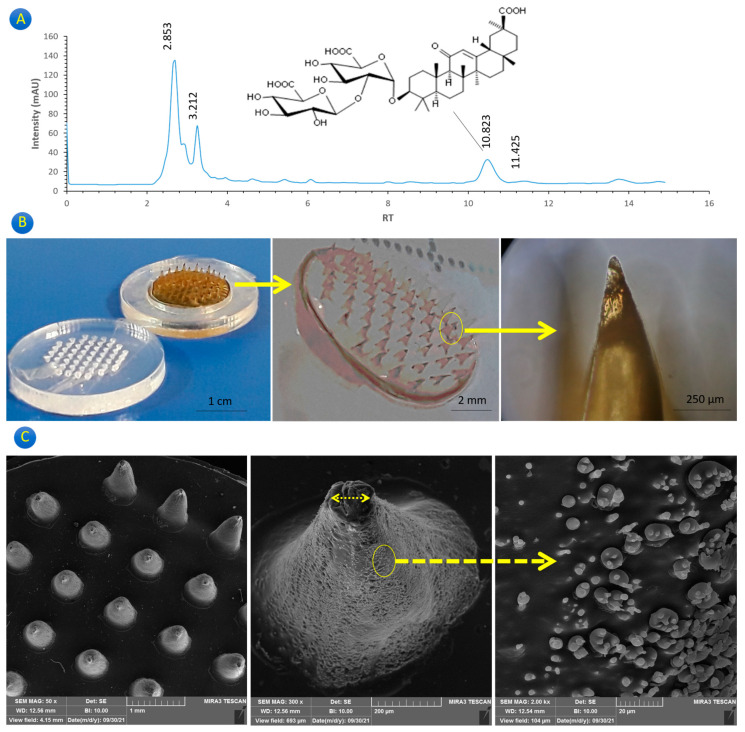
(**A**) HPLC chromatogram of GgE, (**B**) GgE-loaded MNP showing the sharpness level of the created microneedles, (**C**) scanning electron microscopic images of the GgE-loaded MNP.

**Figure 2 gels-10-00087-f002:**
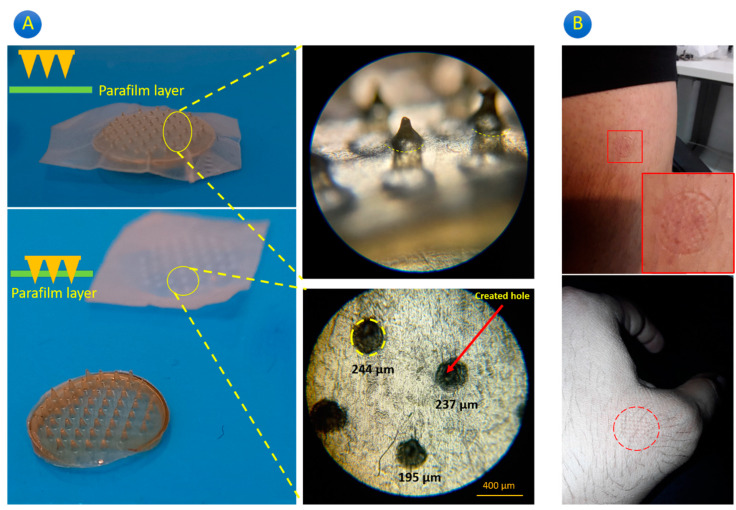
Penetration test using (**A**) parafilm and (**B**) human skin.

**Figure 3 gels-10-00087-f003:**
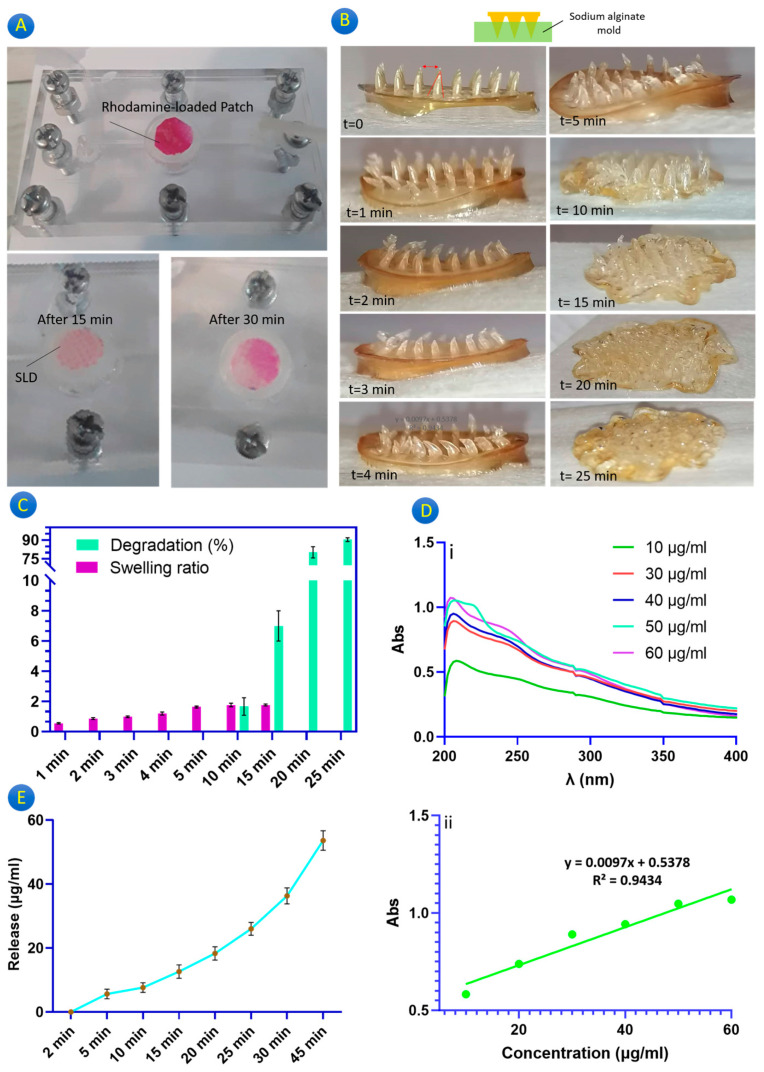
(**A**) Rhodamine release in SLD matrix evaluation using a microfluidic system, (**B**) visual approach to monitor swelling of microneedles, (**C**) swelling and degradation behavior of the GgE-loaded MNP, (**D**) release standard curve, (**E**) release profile of GgE from GgE-loaded MNP.

**Figure 4 gels-10-00087-f004:**
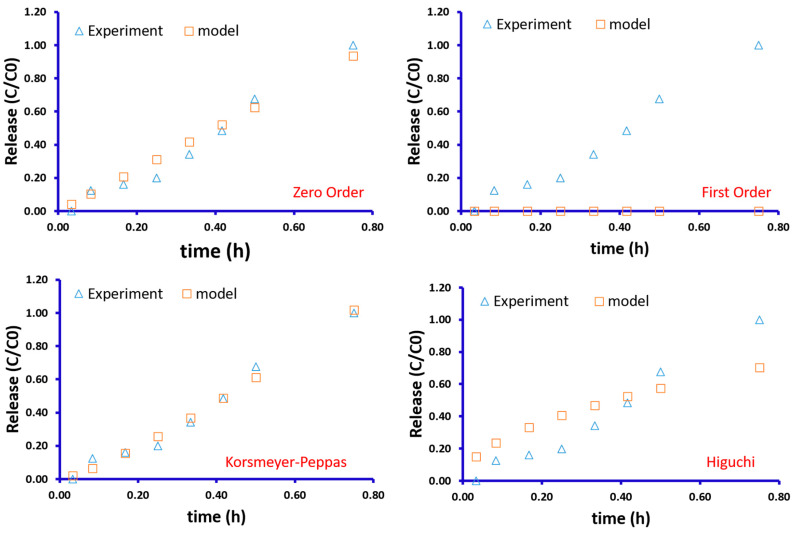
Comparing the experimental results with the obtained results from theoretical models.

**Figure 5 gels-10-00087-f005:**
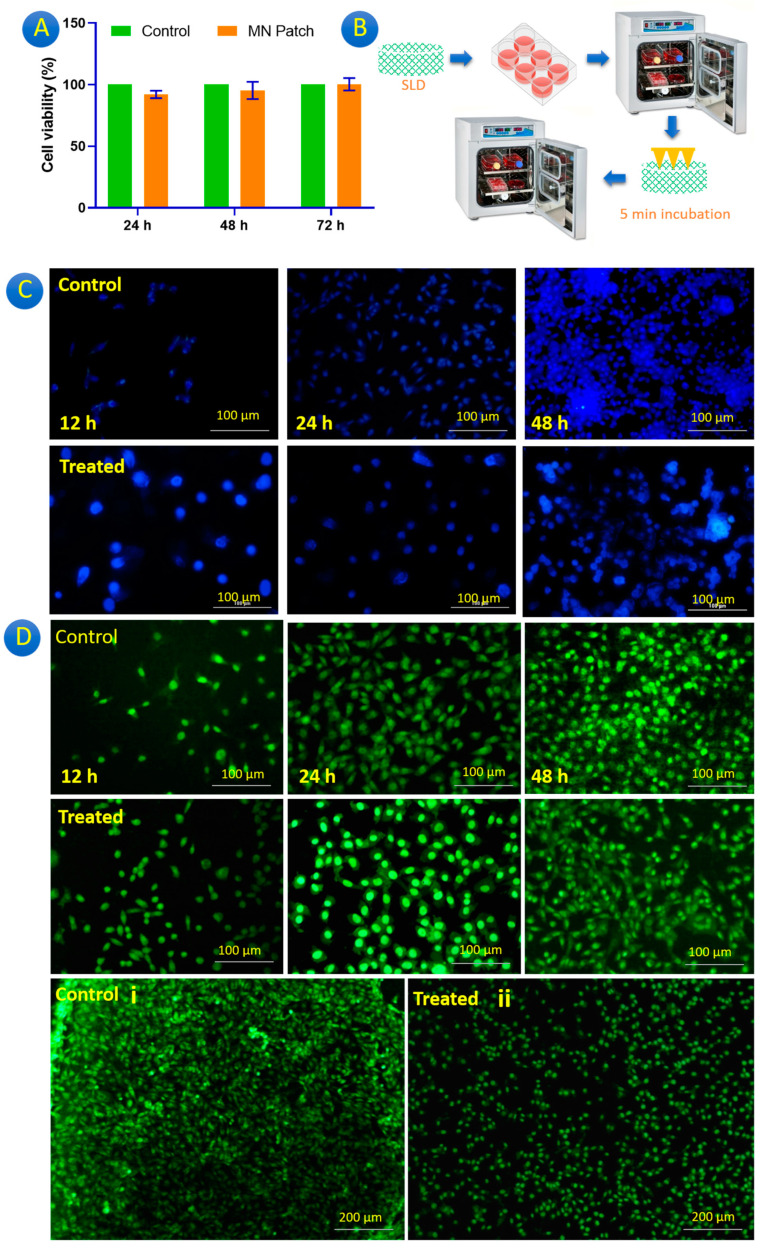
(**A**) Cytotoxicity results for the MNP without GgE, (**B**) schematic procedure for testing GgE-loaded MNP, (**C**) fluorescence micrographs of the DAPI-stained L929 cells, and (**D**) fluorescence micrographs of the acridine orange-stained L929 cells on SLD for control and treated SLDs.

**Figure 6 gels-10-00087-f006:**
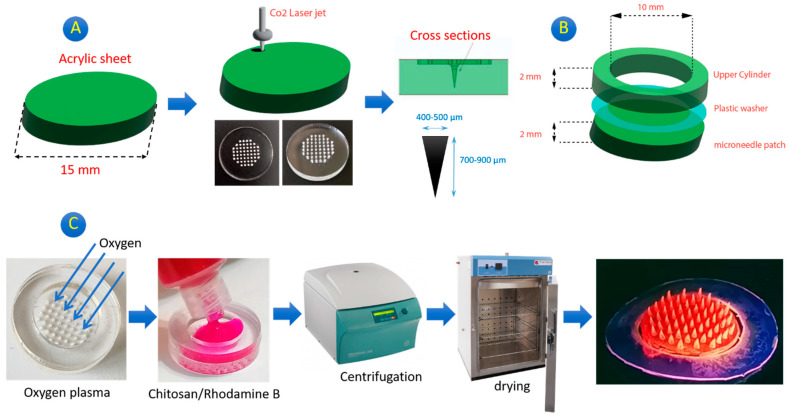
(**A**) The schematic protocol to create a microneedle mold, (**B**) components of the microneedle mold, and (**C**) the schematic process of creating MNP (preparing rhodamine-loaded MNP).

## Data Availability

The data presented in this study are available on request from the corresponding author. The data are not publicly available due to ethical.

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
