# Peer review of "In Vitro Release of Glycyrrhiza Glabra Extract by a Gel-Based Microneedle Patch for Psoriasis Treatment"

_gels, 2024, doi:10.3390/gels10020087_

Round 1

Reviewer 1 Report

Comments and Suggestions for Authors

The manuscript is about the development of microneedle patches that can release therapeutics called Glycyrrhiza glabra extract (GgE) in a slow manner for the treatment of skin-related disorders. Chitosan-based microneedles were prepared and characterized in detail, their cell behavior was investigated in terms of cytotoxicity and cell reduction. Here are some points to pay attention more which might be useful to fix for the readers:

-Figure 2A and 6 have no scale bars.

-In Figure 4E, it would be better if the released amount is represented as its percentage to the loaded amount with respect to time. 

-It is not very clear how alginate mold and chitosan microneedle are peeled off, if there is an electrostatic interaction between the layers? 

-For the semi-skin preparation, more explanation seems to be needed to understand the permeability property, like its similarity to natural akin in terms of pore size? 

Comments on the Quality of English Language

Minor editing of English language required

Author Response

Dear Reviewer

Thanks for the comments. we did our best to answer the comments.

Reviewer 1:

Comments and Suggestions for Authors

The manuscript is about the development of microneedle patches that can release therapeutics called Glycyrrhiza glabra extract (GgE) in a slow manner for the treatment of skin-related disorders. Chitosan-based microneedles were prepared and characterized in detail, their cell behavior was investigated in terms of cytotoxicity and cell reduction. Here are some points to pay attention more which might be useful to fix for the readers:

-Figure 2A and 6 have no scale bars.

Thanks for the comment. Done.

-In Figure 4E, it would be better if the released amount is represented as its percentage to the loaded amount with respect to time. 

Thanks for the comment. Done.

-It is not very clear how alginate mold and chitosan microneedle are peeled off, if there is an electrostatic interaction between the layers. 

Thanks for the comment. Alginate film was used as a semi-skin just to test the penetration. At this step, cross-linked sodium alginate film was used as a hydrogel (holding water) and had a soft structure, while the microneedle patch was used in its dry form. The process was simple. We just put the patch on the alginate film and then detached it. There was no interaction between them.

-For the semi-skin preparation, more explanation seems to be needed to understand the permeability property, like its similarity to natural skin in terms of pore size? 

Thanks for the comment. Considering the semi-skin preparation, it was just used to measure drug release in the chip and approve that the patch has the ability to release the drug. By the way, we added some explanations to make this issue more clear (page 8).

Reviewer 2 Report

Comments and Suggestions for Authors

The paper “In vitro release of Glycyrrhiza glabra extract by a fabricated gel-based microneedle patch to lower cell concentration: a hope for psoriasis” is focused to study to  the potential of the Glycyrrhiza glabra extract on slowing down the rate of cell proliferation using  new microneedle patch.

The paper is interesting, well organized, but a check of English language and grammar should always be done.

Comments on the Quality of English Language

The paper is interesting, well organized, but a check of English language and grammar should always be done.

Author Response

Dear reviewer

thanks for the comments. we did our best to answer all comments.

Reviewer 2:

The paper “In vitro release of Glycyrrhiza glabra extract by a fabricated gel-based microneedle patch to lower cell concentration: a hope for psoriasis” is focused to study to  the potential of the Glycyrrhiza glabra extract on slowing down the rate of cell proliferation using  new microneedle patch.

The paper is interesting, and well organized, but a check of the English language and grammar should always be done.

Thanks for the comment. That is kind of you. We did our best to make the paper more acceptable from the viewpoint of grammar.

Reviewer 3 Report

Comments and Suggestions for Authors

Title

-The authors include psoriasis in the title and it does not have any relationship with the manuscript.

Abstract

-English must be improved. Examples: "it was tried", "Skin disorders"

-What is the cell line L929?

-What is the meaning of this comment?: "the fabricated MNP can be a useful recommendation for in-vitro studies". Is it not applied for in vivo applications?

Introduction.

-Authors should focus on the different existing type of needles and less in ski disorders since the MNP is not applied to cure any specific skin disorder

-There is not enough scientific information related to Glycyrrhiza glabra. Authors should include more information.

Material and methods.

-Authors should show characterization of Glycyrrhiza glabra extract.

-Auhtors should explain the use of C20A4 Human Chondrocyte Cells. The final application is skin?

Results and discussion

-Authors should explain the meaning of "However, this study aimed at the fabrication of low-cost MNP useful for in-vitro studies"

-Is there any regulatory related to this experiment? "The penetration was
also tested on real human skin (two soft and hard parts)"

-Last paragraph: biology mechanism is explained but is it necessary? Since the experiments are preliminary?

Comments on the Quality of English Language

English must be improved along the manuscript

Author Response

Dear Editor

thanks for the comments. we did our best to answer them and hope they are satisfactory.

Reviewer 3:

Title

-The authors include psoriasis in the title and it does not have any relationship with the manuscript.

Thanks for the comment. The title was edited.

Abstract

-English must be improved. Examples: "it was tried", "Skin disorders"

Thanks for the comment.

-What is the cell line L929?

Thanks for the comment. L-929 is an adherent type of mouse fibroblast cell line that was known as an alternate test system for toxicity assessment.

-What is the meaning of this comment?: "the fabricated MNP can be a useful recommendation for in-vitro studies". Is it not applied for in vivo applications?

Thanks for the comment. Actually, we meant that this type of MNP can be a good and non-expensive candidate for lab-scale studies. In some countries like Iran purchasing such equipment (like microneedle mold) is not accessible and too expensive to import.

Introduction.

-Authors should focus on the different existing type of needles and less on skin disorders since the MNP is not applied to cure any specific skin disorder

Thanks for the comment. This study has two purposes, the first and more important is introducing Glycyrrhiza glabra extract as an agent or factor to lower the rate of cell proliferation which is necessary for some skin disorders like psoriasis. The second one is that, since microneedle patches are a good candidate to treat skin disorders, we tried to employ a MNP system to make it more reliable. The main focus of this study is on lowering cell concentration and then recommending a microneedle patch manufacturing to deliver this agent.

We added some information about MNP and their types:

Medical devices with a micron-scale called microneedle patches are used to administer drugs. To give vaccinations, medications, and other therapeutic substances, they are made up of small needles that pierce the skin. The usage of microneedles has expanded beyond transdermal drug administration to include intraocular, vaginal, transungual, cardiac, vascular, gastrointestinal, and intracochlear delivery [13]. Small molecular medications, macromolecular pharmaceuticals (proteins, mRNA, peptides, vaccines), and hydrophilic or hydrophobic drugs can all be delivered via microneedles.

There are several types of microneedle patches used for drug delivery and other applications including Solid Microneedles, Coated Microneedles, Dissolving Microneedles, Hollow Microneedles, and Hydrogel Microneedles. It's vital to remember that depending on the individual application and preferred drug delivery mechanism, several sorts of microneedles may have distinct benefits and disadvantages. To increase the effectiveness of medication administration and the patient experience, researchers are still investigating and creating new kinds of microneedle patches.

-There is not enough scientific information related to Glycyrrhiza glabra. Authors should include more information.

Thanks for the comment. We liked this and you were right. We did the HPLC and added some scientific aspects of Gg (Fig.2A)

Material and methods.

-Authors should show characterization of Glycyrrhiza glabra extract.

Thanks for the comment. We liked this and you were right. We did the HPLC and added some scientific aspects of Gg (Fig.2A)

-Auhtors should explain the use of C20A4 Human Chondrocyte Cells. The final application is skin?

Thanks for the comment. Sorry for this mistake. we used our previous protocol and forgot to change the cell line. As previously mentioned, the used cell line was L929 which is for skin.

Results and discussion

-Authors should explain the meaning of "However, this study aimed at the fabrication of low-cost MNP useful for in-vitro studies"

Thanks for the comment. This sentence was removed.

-Is there any regulations related to this experiment? "The penetration was also tested on real human skin (two soft and hard parts)"

Thanks for the comment. If we understood your comment correctly, we meant that the fabricated patch has the potential for both in vivo and in vitro studies.

-Last paragraph: biology mechanism is explained but is it necessary? Since the experiments are preliminary?

Thanks for the comment. You are right and we do agree. Please consider this part as a discussion and a hypothesis. We just tried to give some hypothetical information based on the previous study to give some clues for the next studies.

Reviewer 4 Report

Comments and Suggestions for Authors

1- Could you provide more details on the fabrication process of the chitosan-based microneedle patch (MNP) using a CO2 laser cutter? Specifically, what were the key steps involved?

2-You mentioned that the microneedles (MNs) were conical with specific dimensions (diameter: 400–500 μm, height: 700–900 μm). What was the rationale behind choosing these dimensions, and how do they compare to other existing MNs in the literature?

3-The MNs exhibited significant swelling (2 folds) within 5 minutes and good degradability within 30 minutes. Could you elaborate on how this behavior was achieved and what implications it has for drug release kinetics?

4-Can you provide more details about the methods used to assess cytotoxicity against the fibroblast cell line L929? Were there any specific concentrations or exposure times tested?

5-What were the specific findings from the AO and DAPI staining with regard to the reduction in cell population after GgE delivery? Were there any observations on the mode of action or mechanisms involved?

6- The authors can use the following article to improve the discussions of the release, swelling, and introduction descriptions: https://doi.org/10.3390/polym15092031

Author Response

Dear Reviewer

Thanks for the comments. We did our best to follow your comments and make them. We hope you accept the corrections. However, we believe that our paper is now more potent than before.

Best Regards

1. Could you provide more details on the fabrication process of the chitosan-based microneedle patch (MNP) using a CO2 laser cutter? Specifically, what were the key steps involved?

         Response: Thanks for the comment. Done.

In section 2.3.1, we explained more about the fabrication of the mold. We first created a mold and then by using that mold, we fabricated the chitosan-microneedle pach.

To make the mold:

Using the CO2 laser, the laser beam was adjusted on the surface of the acrylic disk (a bulk disk with a diameter of 15 mm. The laser power and engraving speed were the main parameters to be optimized for desirable engraving and almost the same depth per run. The engraving was performed by a simple laser shot on the selected points designed by CorelDraw. The depth of the created cone could be easily adjusted by changing the power and engraving speed. Increasing the power and the rate of engraving resulted in a too-depth cone (> 1 mm).

Power is a crucial laser parameter since it determines the quantity of energy is used throughout the procedure and maybe even how deep the cut will be. The time interval between a single laser pulse's beginning and ending is referred to as pulse width. Higher cutting speeds and improved laser cutting quality are made possible by using the proper pulse widths. The ideal pulse width varies depending on the workpiece material.

To make the chitosan-based patch:

. 0.5 ml of the chitosan/GgE solution was dripped over the mold and allowed to cover the surface of the mold. Then, the mold was centrifuged (using specific holders) at 4000 rpm for 10 min. The centrifuge plays a vital role in the fabrication of the patch with sharp microneedles. Next, the mold was dried at 40 ˚C overnight. Finally, the patch was detached and stored at 4 ˚C.

2. You mentioned that the microneedles (MNs) were conical with specific dimensions (diameter: 400–500 μm, height: 700–900 μm). What was the rationale behind choosing these dimensions, and how do they compare to other existing MNs in the literature?

Response: Thanks for the comment. Done.

The micro-needles are too tiny to see from the naked eye, in fact, less than one millimeter in height. In general, microneedles' dimensions vary from 50 to 900 micrometers in length or height and have a diameter of 1 millimeter (a). The designed MNs must be able to penetrate the skin deeply without breaking. MNs ought to be the ideal size.

The rationale behind choosing these dimensions turns back to the reality about microneedles as if they must be tiny to cause less pain.

The length (high) of the microneedle and its diameter are the main parameters to compare them. There are numerous published paper focusing on microneedle patches with distinct dimensions.

a: https://scholar.google.com/scholar_lookup?journal=Pharmaceutics.&title=Transdermal+delivery+of+drugs+with+microneedles%E2%80%94potential+and+challenges&author=K+Ita&volume=7&issue=3&publication_year=2015&pages=90-105&pmid=26131647&doi=10.3390/pharmaceutics7030090&

3. The MNs exhibited significant swelling (2 folds) within 5 minutes and good degradability within 30 minutes. Could you elaborate on how this behavior was achieved and what implications it has for drug release kinetics?

Response: Thanks for the comment. Done.

Polymers and hydrogels' ability to release drugs can be significantly impacted by swelling. A key characteristic of hydrogels that affects drug release rates via regulating drug diffusion is their capacity to swell. Because of the connection between the swelling and the drug concentration as well as the presence of the swelling convective term, swelling dynamics have a significant impact on drug transport.

“The effect of swelling on drug release can be significant and should be carefully considered in pharmaceutical formulations. When a drug delivery system comes into contact with a liquid, it may undergo swelling due to the absorption of the liquid by the polymer matrix. This swelling can affect the release of the drug from the delivery system. One important aspect to consider is the swelling kinetics. Different polymers have different swelling characteristics, and their ability to take up liquid can vary greatly. It is crucial to understand the kinetics of swelling for the specific polymer used in the drug delivery system. This can be determined experimentally by measuring the change in size or weight of the polymer as it absorbs the liquid [24]. When the microneedles penetrate the skin, they induce a local immune response, leading to swelling. This swelling can potentially affect the release of drugs from the microneedles. Besides, microneedle swelling can also modify the microenvironment around the microneedles. The increased swelling can lead to changes in the local pH, temperature, or moisture content. These changes can influence the solubility and stability of the drug molecules, further impacting their release [25]. It is important to note that the effect of microneedle swelling on drug release can vary depending on several factors, including the composition of the microneedles, the properties of the drug, and the specific application.

Understanding the many elements that may cause medication degradation in microneedle patches is crucial. These variables include, among others, temperature, pH, and humidity. In terms of drug release, degradation can significantly impact the release profile of the drug from the patches [27]. As the polymeric matrix degrades, the polymeric networks may change, which can affect its solubility and release kinetics. Therefore, microneedles gradually release drugs during their slow degradation under the skin. It highlights the point that the performance of a microneedle patch can be affected by the degradation of the carrier matrix, which is directly linked to drug diffusion, dissolution, and degradation.”

4. Can you provide more details about the methods used to assess cytotoxicity against the fibroblast cell line L929? Were there any specific concentrations or exposure times tested?

Response: Thanks for the comment. Done.

the patches were seeded with 10000 fibroblast cell line L929 cells/well in a 96-well plate.

the patches were incubated for 48 h at 37°C and 5% CO2

5. A) What were the specific findings from the AO and DAPI staining with regard to the reduction in cell population after GgE delivery? B)Were there any observations on the mode of action or mechanisms involved?

            Response: Thanks for the comment. Done.

We used these techniques to monitor cells concentration and their death. They detect cells by staining their specific components like DNA and cytoplasm. After GgE delivery, these staining showed that the cell population reduced without a specific death which indicates the reduction in cell proliferation.

One of the purposes of DAPI staining is to study the cell cycle, determine the index of mitosis in an organism, or count cells [33]. DAPI staining results revealed that both groups' cell populations expanded properly, with distinct cell nuclei and no discernible dead cells  (Fig. 5C). In the control group, Cells seemed to be attached to the SLD and with higher concentration after 48h in contrast to 24h. Based on Fig. 5C, the GgE ex-tract delivery reduced cell proliferation compared to the control group. This finding indicates that in skin disorders like Psoriasis in which cells have non-stop proliferation, a transdermal delivery system like the GgE-extract microneedle patch can reduce the growth of cells and control this disease.

A) Studying the mechanism was out of the main concerns of this study due to the defined proposal. However, we used this wise comment as a recommendation for future perspectives.

6. The authors can use the following article to improve the discussions of the release, swelling, and introduction descriptions: https://doi.org/10.3390/polym15092031

Response: Thanks for the comment. Done. We used this reference in the discussion about swelling and degradation(Ref. 26)

Reviewer 5 Report

Comments and Suggestions for Authors

The authors reported a gel-based microneedle patch containing a naturally occurring drug for transdermal purposes. Generally speaking, bioavailability and percutaneous behavior are of high importance for the therapy. Microneedle is a good method for loaded drugs to overcome the skin barrier function and improve drug penetration across skin and bioavailability. The authors conducted the experiments to separately evaluate the penetration behavior by using parafilm and real human skin, and release behavior by using SLD. Actually, a better method is to use Franz diffusion cells in combination with animal skin like pig ear skin, etc. The data would be much more convincing.  The following concerns should also be addressed before being accepted for publication. 

1. in Figure 4E, from your data, the release saw a clear increase as time went on. so how about 50 mins, 60 mins, and so on?  So it'd be better to use accumulative release to characterize the release profile, and you can get the final release percentage. Theoretically, it would be flattening in the end. 

2. in 2.9 biodegradability, what are your experimental conditions? similar to 2.8? 

3. in part 3 Results and Discussion, it'd be better to use subheads to navigate your discussion and make it clear. 

4. for biosafety and cytotoxicity, the normal practice is to use well-accepted standard methods like ISO 10993-5. 

5. The full name of DAPI should be given at its first appearance. 

6. in P6, the authors discussed the major active components in the herbal extract. References should be given to support your argument, like "It has been discovered to ....".

7. below Figure 4, there is a standalone sentence "...in the case of insertion....", what do you mean? 

8. the authors mentioned Fig 1S. where is it?

Comments on the Quality of English Language

There are some language errors in the manuscript. Generally speaking, the past tense should be used to describe what you have done, instead of present tense. 

in P10, paragraph 3, "it has been decided that the release study be carried out...", it's a grammar error. 

in P12 paragraph 4, "to answer this more biological...which this issue...", it's also wrong grammatically. 

Author Response

Dear Reviewer

Thanks for the comments. We did our best to follow your comments and make them. We hope you accept the corrections. However, we believe that our paper is now more potent than before.

Best Regards

  1. in Figure 4E, from your data, the release saw a clear increase as time went on. A) so how about 50 mins, 60 mins, and so on?  B) So it'd be better to use the accumulative release to characterize the release profile, and you can get the final release percentage. Theoretically, it would be flattening in the end. 

Response: Thanks for the comment. What a wise comment.

A) We tried to work on the drug release according to the degradation time. However, we did the research for 45 min, while after 25 min nearly 90% of the patch was degraded.

B) What a wise comment. It was great and we are happy with this comment because it gave us more ideas. We followed your order. We also studied release kinetic models including first order, zero order, Korsmeyer-Peppas, and Higuchi models to find out the most appropriate model to predict the results. Please check section 3.6.

     2. in 2.9 biodegradability, what are your experimental conditions? similar to 2.8? 

Response: Thanks for the comment. Done. We added the required information.

     3. in part 3 Results and Discussion, it'd be better to use subheads to navigate your discussion and make it clear. 

Response: Thanks for the comment. Done.

     4. for biosafety and cytotoxicity, the normal practice is to use well-accepted standard methods like ISO 10993-5. 

Response: Thanks for the comment. We followed a standard protocol for this technique which has been established in our lab. The MTT assay is a common and well-known technique that is used in all studies considering cell-viability. We mentioned the standard number.

    5. The full name of DAPI should be given at its first appearance. 

Response: Thanks for the comment. Done.

DAPI (4',6-diamidino-2-phenylindole)

    6. in P6, the authors discussed the major active components in the herbal extract. References should be given to support your argument, like "It has been discovered to ....".

Response: Thanks for the comment. Done.

“…The most important one is related to glycyrrhizic acid (known as Enoxolone (ENx)). It has been discovered that Enoxolone is a pentacyclic triterpenic organic acid [17]. Licorice root also contains other phytocompounds such as glycyrrhizin, 18βglycyrrhetinic acid, glabrin A and B, and isoflavones [18]. However, it was also reported that Enoxolone can be extracted from biological materials including rat blood, human urine, and rat bile [19]. It can be concluded that this agent exists in the human body and no side effect can be hypothesized for its release under transdermal delivery approaches.”

   7. below Figure 4, there is a standalone sentence "...in the case of insertion....", what do you mean? 

Response: Thanks for the comment. Done.

“…To make this issue more clear, the rhodamin-loaded MNP is dry, while the SLD is saturated with water, and after inserting the MNP in the SLD the microneedles start water adsorption and swelling. the penetrated water dissolves Rhodamin and due to the difference in Rhodamin concentration inside and outside of the microneedles, Rhoda-mins diffuses out of the microneedles (inside the SLD). In the case of insertion in human skin, a similar mechanism happens due to the skin interstitial fluid.”

   8. the authors mentioned Fig 1S. Where is it?

Response: Thanks for the comment. Done. It has been deleted. There is no Fig.1S.

Round 2

Reviewer 5 Report

Comments and Suggestions for Authors

Some minor errors like "CO2 laser" (the subscript) can be found. Check your manuscript carefully.